# Exploring Critical Factors Associated with Completion of Childhood Immunisation in the Eastern Province of Saudi Arabia

**DOI:** 10.3390/vaccines10122147

**Published:** 2022-12-14

**Authors:** Marwa Alabadi, Tawfiq Alashoor, Omran Aldawood, Zainab Qanbar, Zakariya Aldawood

**Affiliations:** 1School of Nursing and Midwifery, Faculty of Health and Medicine, University of Newcastle, Callaghan, NSW 2308, Australia; 2Department of Digitalization, Copenhagen Business School, Howitzvej 60, 2000 Frederiksberg, Denmark; 3Primary Health Care Division, Ministry of Health, Riyadh 12271, Saudi Arabia; 4Primary Health Care Division of Qatif City, General Directorate of Health Affairs in the Eastern Region, Ministry of Health, Qatif 31911, Saudi Arabia

**Keywords:** public healthcare, childhood immunisation, childhood vaccination, vaccine hesitancy, delayed vaccination, Saudi Arabia

## Abstract

(1) Background: surveillance data from the Saudi Ministry of Health shows that the Kingdom’s large-scale immunisation programme has significantly reduced the mortality and morbidity of the target diseases among children. In this study, we review relevant literature and test a number of hypotheses related to the association between demographic, socio-economic, clinic-related, and parents-related variables and completion of childhood immunisation. In doing so, this study identifies critical factors associated with completion of childhood immunisation and presents important implications to healthcare practitioners, particularly in Saudi Arabia; (2) Literature review: a systematic literature review was conducted to understand what is currently published concerning parents’ immunisation compliance in Saudi Arabia and the factors associated with immunisation compliance. (3) Methods: from March to May 2022, an online survey was administered to parents attending one of the 27 primary health care (PHC) centres in Qatif. Data from parents (n = 353) were analysed using exploratory factor analysis, correlation, and a series of OLS and logistic regression models; (4) Results: parental (child) age was negatively (positively) associated with the completion status of childhood immunisation (both *p* < 0.05). Parents with positive attitudes, social norms, perceptions towards immunisation, and those working in private companies were more likely to immunise their children (all *p* < 0.05). Conversely, living in an apartment building, walking to PHCs, waiting longer at PHCs, and having higher knowledge of immunisation were negatively associated with the completion of childhood immunisation (all *p* < 0.05); (5) Conclusions: several factors that positively or negatively influence the completion of childhood immunisation have been identified. Future studies may investigate the causal link between these factors and parental decision-making regarding childhood immunisation.

## 1. Introduction

Immunisation is an effective strategy for eliminating infectious diseases, preventing approximately 4–5 million deaths worldwide annually [1]. As immunisation is considered one of the most cost-effective public health interventions to reduce the mortality and morbidity of diseases, it is crucial to expand access to immunisation and achieve the World Health Organisation (WHO) Good Health and Well-Being—Sustainable Development Goal (SDG) by 2030 [2]. Such a strategy is specifically important for developing countries with poor health infrastructure and limited access to health care; however, it is also essential for countries experiencing an increase in vaccine non-compliance.

Children are specifically vulnerable to acquiring infectious diseases due to their untrained and undeveloped immune systems, highlighting the importance of immunising children against infectious diseases [3,4,5]. As a result, many countries around the world conduct immunisation campaigns targeting common childhood infectious diseases. An estimated 116.3 million (approximately 86%) children under the age of one year received the completed immunisation, inclusive of three doses, for diphtheria-tetanus-pertussis (DTP3) globally [6].

Saudi Arabia is a developing country whose full immunisation programme has been implemented since 1984 as an essential and integrated element of PHC. This immunisation programme has significantly reduced mortality and morbidity among children from the target diseases in Saudi Arabia [7]. Yet, similar to other countries, Saudi Arabia faces the challenge of an uneven immunisation completion status across its population due to limited health access in remote areas, moderate literacy rates of parents, and social stigmas [8]. While childhood immunisation non-compliance has been reported in every country, developing countries show a high prevalence of non-completion rates of child immunisation [9]. Completion of childhood immunisation in Saudi Arabia varies across provinces, with some areas reporting up to 20% of immunisation non-compliance [10]. Given the dearth of research in this area and particularly in Saudi Arabia, it is important to explore immunisation compliance rates across urban and rural areas of the country, as well as the factors driving non-compliance.

Generally speaking, the responsibility to vaccinate children lies with their parents. This responsibility makes parents’ knowledge and attitudes towards immunisation a very important factor for the protection of their children from vaccine-preventable diseases [11]. Previous research identified several factors associated with poor compliance with immunisation by parents [12]. This includes mistrust of and feeling alienated by the paediatrician, which reduces mothers’ trust in immunisation [13], fear of the side effects of immunisations [14], religious and anti-government sentiments, and a belief in the harmless nature of diseases [15]. Similarly, other studies [13] identified mistrust in the efficacy of vaccines, lack of insurance, disease irrelevance, and mistrust in governments as factors contributing to immunisation non-compliance. In a large country like Saudi Arabia, with vast uninhabited lands, socioeconomic and geographical variables may also account for variations in immunisation rates between regions. For example, residents of a remote region of the country with poor access to health care facilities may have lower compliance rates and are likely to be less educated and harbour more misconceptions about immunisations, which can negatively influence their decisions to vaccinate their children [16]. 

Surveillance data from the Saudi Ministry of Health shows that the Kingdom’s large-scale immunisation programme has significantly reduced the mortality and morbidity of the target diseases among children [17]. In this study, we review relevant literature and test a number of hypotheses related to the association between demographic, socio-economic, clinic-related, and parents-related variables and childhood immunisation completion rate. In doing so, this study contributes to the identification of critical factors associated with completion of childhood immunisation and presents important implications to healthcare practitioners, particularly in Saudi Arabia.

Recent studies have described several factors that impact the national childhood immunisation compliance rate in Saudi Arabia [8,11,18,19]. These studies have been limited to parents’ experiences from urban areas of Saudi Arabia and hence; a more comprehensive examination of these factors is required in both city and rural dwelling families. Data on the immunisation coverage of children have identified particular sociodemographic characteristics as determinants of the completion status of selected vaccines, but further research is still needed to determine whether these characteristics differ in more rural communities [20]. 

Furthermore, the disruption of services internationally during 2020 and 2021 due to the COVID-19 pandemic has been shown to have increased immunisation hesitancy [21,22]. Immunisation rates are expected to drop further, which may contribute to an increased incidence of vaccine-preventable diseases [23]. Thus, it is important to further explore immunisation compliance and contributing factors to vaccine hesitancy in rural and urban areas of Saudi Arabia. This study contributes to the National Childhood Immunisation Programme by exploring critical factors associated with completion of childhood immunisation and the resulting immunisation compliance across the rural and urban areas of Qatif, located in the Eastern Province of Saudi Arabia. Identifying the factors that can significantly influence completion of childhood immunisation has important practical implications which are dis-cussed in much further detail in the discussion section.

## 2. Literature Review

A literature review was conducted to understand what is currently published concerning parents’ immunisation compliance in Saudi Arabia and the factors associated with immunisation compliance. This review critically discusses and summarises the main findings in this section. Gaps in the literature were identified, the research questions and objectives were formulated, and their justifications were provided.

### 2.1. Literature Review Questions

What are the rates of immunisation compliance in Saudi Arabia?What are the factors that contribute to parents’ immunisation compliance in Saudi Arabia?

#### 2.1.1. Search Strategy

A comprehensive literature search was carried out using five online databases, ‘PubMed’, ‘Google Scholar’, ‘Scopus’, ‘Medline’, and ‘ScienceDirect’. The following search terms, using Boolean operators, were used: ‘childhood immunisation’ OR ‘immunisation’, ‘knowledge’ OR ‘attitude’ OR ‘factors affecting immunisation’ OR ‘compliance’ OR ‘non-compliance’ OR ‘hesitancy’ AND ‘Saudi Arabia’.

#### 2.1.2. Inclusion Criteria

The following inclusion criteria were used: (1) studies including parents, adult guardians, or caretakers who are responsible for the immunisation of their children; (2) studies assessing the knowledge, attitudes, and perceptions of parents, guardians, or caretakers concerning childhood immunisations; (3) studies exploring the childhood immunisation compliance of parents, guardians or caretakers; (4) studies published in English; (5) studies published in the past 10 years; (6) studies conducted in Saudi Arabia.

#### 2.1.3. Exclusion Criteria

The following exclusion criteria were applied: (1) studies investigating the knowledge of parents, guardians, or caretakers about other subjects rather than childhood immunisations; (2) studies investigating vaccine safety; (3) studies investigating influenza and COVID-19 immunisation; (4) studies conducted outside Saudi Arabia; (5) studies published in non-English languages.

#### 2.1.4. Search Outcomes

Initially, 447 studies were found. After removing duplicates, 323 studies were reviewed for relevance by reading the titles and abstracts. Of these, 16 were screened based on the inclusion and exclusion criteria, and eleven studies were excluded because of their incorrect target populations or outcomes. Five studies were identified as relevant due to their relevance and potential significance. The flow diagram of the search scheme is presented in Figure 1.

### 2.2. Findings of the Literature Review

The five studies included in the literature review were all cross-sectional studies. These studies were conducted in Saudi Arabia’s Riyadh (4 studies), Jeddah (1 studies), Taif (1 study), Hail (1 study) and Al-Qassim (1 study) regions. The regions explored in these studies are urban localities with no remote or rural areas. All studies were published in English. A summary of the included studies is detailed in Table 1.

#### 2.2.1. Rates of Immunisation Compliance

The definition of non-compliance was consistent across the three studies that reported immunisation non-compliance [8,18,24]. Non-compliance was defined as the parents’ delay or refusal to allow the administration of a vaccine to a child, despite the availability of a vaccine [25]. The relative proportion of parents who reported being non-compliant with the National Childhood Immunisation Programme in Saudi Arabia was 14.8% (57 out of 384 parents) [8], 17% (51 out of 300 parents) [24], and 20% (100 out of 500 parents) [18]. All three studies were conducted in Riyadh city.

Alshammari et al. reported that 86% of their 467 randomly selected parents from the Hail region completed their children’s mandatory immunisations [16]. Alsubaie et al. noted that 20% of the 500 parents interviewed were hesitant to vaccinate their children, and 36% of those children were not vaccinated fully for their age [24]. A study of 668 parents from Najran reported that only 61.8% complied with the immunisation schedule [20]. This latter finding is seriously alarming. 

#### 2.2.2. Factors That Impact Completion of Childhood Immunisation

The five papers selected for this review discussed various factors influencing childhood immunisation. These primary factors include parental education, travel, availability of vaccines, demographical characteristics, and family size.

**Parents’ knowledge** emerged as a major reason for reduced childhood immunisation compliance and was reported in two studies [18,24]. The main areas identified included the knowledge and perception related to immunisation requirements, immunisation safety, and side effects. In addition, 41% of parents in Taif, Saudi Arabia, believed their children no longer needed immunisation for diseases that had been eradicated in Riyad [26]. Parents believed that making several visits to a health care facility or giving multiple shots to their children was not safe [24]. Mild flu-like symptoms or fever following a previous immunisation also emerged as a factor associated with poor compliance by the parents [26].

**Travel or access** was highlighted by three studies [11,16,27]. Parents who travelled long distances to acquire immunisations for their children were less likely to complete their children’s immunisations [16]. Poor immunisation compliance was also influenced by time constraints on immunisation day or potential travel hazards for their children [11]. Additionally, access to vaccines or parental perception of vaccine unavailability in health care centres was identified in one study as contributing to immunisation non-compliance [27].

**Parents’ demographic characteristics** were identified to play a role in the reluctance to complete childhood immunisation in five studies [8,11,24,26,27]. For example, non-Saudi parents living in Saudi Arabia were more likely to refuse immunisation as compared with Saudi parents [26]. Parental age was correlated with hesitancy, with those aged < 35 years having higher immunisation compliance rates [11]. The non-employment of the father, poor income, and large family size also emerged as potential demographic features contributing to immunisation refusal [27].

**Table 1 vaccines-10-02147-t001:** Brief overview of the studies included for analysis.

Study (Region)	Sample Size	Age of Children	Vaccine of Inquiry	Conclusion
[8] (Riyadh)	384	<14 years	National Childhood Immunisation Programme	Most parents showed adequate confidence in the efficacy of vaccines. Only a tiny proportion showed doubts.
[24] (Riyadh)	500	2 m–7 years		A small proportion of parents showed vaccine hesitancy.
[27] (Jeddah)	351	<3 years		Adherence to immunisation was common but some parents reported delays.
[16] (Hail region)	467	<5 years		Most parents demonstrated a good amount of confidence in and acceptance of vaccines.
[27] (Taif)	731	0–12 years		Most parents had good knowledge of and a positive attitude towards vaccines.

#### 2.2.3. Discussion of the Literature Review

The major factor identified as contributing to childhood immunisation compliance is parents’ knowledge and confidence. Parents showed confidence in the efficacy of single vaccines but not in the efficacy of vaccine combinations [24]. This mistrust can partly be explained by the educational status of the parents, as parents with higher education were more likely to vaccinate their children [16,28]. Another study showed conflicting results highlighting that parents with post-graduate degrees demonstrated higher vaccine hesitancy than those with a bachelor’s or a high school education [24]. While there is no apparent justification for this discrepancy, it is still likely that the educational status of parents might play a role, as parents with a medical background showed a more positive attitude towards immunisation [28].

Local culture, individual experiences, and the influence of pseudoscience websites can impact parental decisions regarding immunisations [29]. Several demographic characteristics also influenced parents’ decisions. Older parents were more likely to vaccinate their children than younger parents [29]. This may be because older parents have likely encountered or seen more of the disease under question than younger parents. Similarly, parents with one or two children were more likely to vaccinate their children. Contrarily, parents with three children or more were less likely to vaccinate their children.

These studies have some limitations. The data collection instruments varied between studies, potentially accounting for some of the discrepancies in the studies. None of the studies were multicentric, and the high risk of bias associated with single-centre studies cannot be ignored. Most of the studies covered the central Saudi Arabian region, so the relatively remote regions of the country were overlooked. One study was conducted with admitted patients only and did not include the healthy population [11]. As a result, hospitalisation may influence parents’ perceptions of their children’s immunisations.

### 2.3. Conclusion of the Literature Review

There is a dearth of literature concerning the factors associated with completion of childhood immunisation in Saudi Arabia. The literature exploring compliance has identified variations between the immunisation rates reported by parents and those reported by the WHO national data [23]. The review of the limited available literature concluded that trust in immunisation is not uniform throughout Saudi Arabia, and a small proportion of parents have doubts about vaccine efficacy. Several factors that influenced parents’ completion of childhood immunisation were reported. The factors and their effects varied between regions, with current studies restricted to urban parts of Saudi Arabia. Thus, several limitations in these studies may have partly accounted for these variations, necessitating a comprehensive investigation into immunisation compliance among parents.

## 3. Conceptual Framework and Hypotheses

Delaying childhood immunisation has become a public health threat. Although immunisation coverage of children in nations remains high, exemption requests for childhood immunisation have increased globally, with a recent drop in immunisation rates [30]. The controversy surrounding the risks and benefits of childhood immunisations has led parents to question their safety [31]. The large amount of conflicting information available online to consumers has been acknowledged as one of the primary barriers to childhood immunisation for parents [32,33]. Thus, to build upon the understanding of parental influence concerning health protective behaviours, this study applies the Theory of Planned Behaviour (TPB) in the context of childhood immunisations.

TPB was developed as an attempt to predict human behaviour [34]. The theory asserts that an individual’s attitude towards a particular behaviour is driven by their evaluation and understanding of the available information [35]. According to the TPB, the three constructs that determine behavioural outcomes are attitudes, subjective norms, and perceived behavioural controls, which influence behavioural intention and hence actual behaviour, as shown in Figure 2 [36].

The first construct is attitude towards the behaviour, which is the extent to which a person has a favourable or unfavourable appraisal of a given behaviour. Attitudes consist of behavioural beliefs and outcome evaluations. The second construct is subjective norms, which reflect the social pressure to perform or refrain from performing a given behaviour. Subjective norms consist of normative beliefs and the motivation to comply. The third construct is perceived behavioural control which refers to people’s perception of the ease or difficulty of performing the behaviour of interest. According to the TPB, these constructs determine behavioural intention, which is the motivational factor that influences actual behaviour [34]. The stronger the intention to engage in a given behaviour, the more likely it is to perform that behaviour.

The theory has demonstrated moderate to high success in predicting a wide range of health behaviours, such as smoking, drinking, substance abuse, and lifestyle modification [36]. The TPB has also successfully identified factors associated with behaviours towards immunisation programmes [37]. Askelson et al. previously assessed mothers’ intentions to vaccinate their daughters against human papillomavirus (HPV) using the TPB [38]. To build upon the limited understanding of how conflicting information affects parental decision-making regarding health protective behaviours, Li et al. extended and applied the TPB in the context of childhood immunisations [33].

The conceptual framework for this study is divided into four categories that have been identified in the literature. As shown in Figure 3, the categories include demographic variables, socioeconomic variables, and clinic-related variables. In addition, the parents-related variable includes parents’ attitudes, subjective social norms, perceived behavioural control, knowledge, perceived risks, and perceived benefits. Based on the pro-posed framework and the TPB, the following hypotheses were posited:


**Demographic Variables**
a.
**Hypothesis 1 (H1):**
*Demographic variables (e.g., age, gender, etc.) are associated with childhood immunisation completion status.*


**Socio-economic Variables**
b.
**Hypothesis 2 (H2):**
*Socioeconomic variables (e.g., place of residence, level of education, etc.) are associated with childhood immunisation completion status.*


**Clinic-related Variables**
c.
**Hypothesis 3 (H3):**
*Clinic-related variables (e.g., distance to PHC centre, waiting time at immunisation clinic, etc.) are associated with childhood immunisation completion status.*


**Parents-related Variables**
d.Theory of Planned Behaviour
**Hypothesis 4 (H4):***A positive attitude toward immunisation is associated with childhood immunisation completion status.***Hypothesis 5 (H5):***Subjective social norms toward immunisation are positively associated with childhood immunisation completion status.***Hypothesis 6 (H6):***Perceived behavioural control toward immunisation is positively associated with childhood immunisation completion status.*
e.Knowledge**Hypothesis 7 (H7):***Higher knowledge about immunisation is positively associated with childhood immunisation completion status.*f.Trade-offs (Perceived Benefits and Risks) **Hypothesis 8 (H8):***Higher perceived risks of immunisation are negatively associated with childhood immunisation completion status.***Hypothesis 9 (H9):***Higher perceived benefits of immunisation are positively associated with childhood immunisation completion status.*


## 4. Methodology

### 4.1. Study Area

The study was conducted in Qatif, which is located in the Eastern Province of Saudi Arabia. Qatif has an area of 611 km^2^ and a population of 1,100,000. In Qatif, 27 public health centre clinics provide childhood immunisations: 20 are located in urban areas and 7 in rural areas. Qatif Regional Public Hospital is located about 5–10 km from the rural areas in Qatif. Qatif does not have public transport infrastructure in rural areas to support transport to urban areas. Therefore, the PHCs provide a range of services to local families, including immunisations, maternal and child health care, community services, mental health, rehabilitation, and health and nutrition education.

### 4.2. Study Design and Sample Size Determination

A statistical power analysis was conducted to determine the sample size. Based on Qatif’s population of 1,100,000 and the need to keep the confidence interval as 95%, the margin of error as 5%, and population proportion as 50%, we need around 350 parents visiting 27 selected PHC clinics to obtain meaningful data. The population of interest consists of parents (guardians or carers) presenting their children (two years old or younger at the time of data collection) to one of the 27 PHC clinics across Qatif.

An online questionnaire considering all three constructs within the TPB was adopted from a previous study conducted in western Saudi Arabia with some modifications [29]. The questionnaire explored parental knowledge, perceived risks, and perceived benefits of vaccinating their children. It also accounted for several variables, potentially affecting decision-making in a region-specific manner. The questionnaire was structured into five main sections: (1) demographics, (2) socioeconomic characteristics, (3) clinic-related variables, (4) parents-related variables, and (5) completion status of childhood immunisation. The first two sections (demographic and socioeconomic characteristics) included questions concerning parents age, gender, employment status, income, education, family size, place of residence, etc (see Table 2). A 7-point semantic differential scale and a 7-point Likert-type scale were used for the third and fourth sections (clinical and parents-related variables), respectively. Lime Survey platform was used for the online questionnaire, which enabled us to offer the same questionnaire in two different languages (Arabic and English). In addition, the online platform enabled participants to switch languages at any time during the survey without restarting their participation.

### 4.3. Recruitment and Data Collection

This study obtained ethical approval from the University of Newcastle, Australia (ethics reference no. H-2021-0378), and the Ministry of Health of Saudi Arabia (ethics reference no. QCH-SREC07/2022). The Director of Scientific Research in the Ministry of Health of Saudi Arabia sent an official letter to all 27 PHC clinics to support the collection of the required data. During the study, precaution was taken to ensure that the four ethical principles of beneficence, non-maleficence, respect for autonomy, and justice were addressed and met.

During the months of March 2022 and May 2022, all 27 PHC clinics advertised the study with a flyer in the waiting area. A Participant Information Statement (PIS) was also placed on the reception desk and was provided to parents who attended the PHC. A survey hyperlink and a QR code were provided in the flyer and PIS. When a potential participant accessed the survey link, the PIS was provided on the first page. After reading the PIS, they would “click to start” to provide their implied consent.

The study included 353 participants in total. Two participants had many missing values and were excluded from the final analysis, resulting in a final sample size of 351 observations. Most of the participants indicated that they completed the childhood immunisation (70.30%). The majority of the participating parents were females (55.00%). The average age of the participating parents and the other parents was 33.65 and 33.80 years, respectively. Forty-five percent of the children were female, varying in their order in the family (see Table 2).

**Socioeconomic variables** identified that the average number of children in a family was 2.72. Approximately 86% of the males responding had a university degree or higher, while 72% of the females reported having a university degree. Thirty four percent of the fathers were government employees, while 59% were private sector employees. On the other hand, approximately 32% of the mothers were government employees, while 18% were private sector employees. Additionally, 50% of mothers reported being neither employed nor students, compared to only 7% of fathers. The average number of people living in a household was five. Most participants revealed that they live in an apartment building (66%), while 32% indicated that they live in a villa. Only 2% of the participants reported living in a mud house or similar structure. Most participants (64.30%) were homeowners, compared to 35.70% who rented accommodation.

**Clinic-related variables**, the average distance that participants travelled to PHC centres was 2 kilometres. Most of the participants reported that they commute to PHC clinics. The majority of parents indicated that they waited 15 to 30 min in the clinic for the child immunisation. The majority (62%) of participants identified that their PHC sent reminders when their next immunisation was due. When asked to rate the immunisation clinic facilities on a scale from 0 to 4 (poor; fair; good; excellent), on average participants rated the facilities as fair-good (2.52).

**Parents-related variables**, Table 2 includes the participating parent’s score on attitude, social norms, behavioural control, knowledge, perceived risks, and perceived benefits. Table 3 shows a list of the PHC clinics the participants are associated with where they get their children vaccinated.

### 4.4. Method of Data Analysis

The quantitative data analysis was conducted using Statistical Package for the Social Sciences (SPSS) software version 26. For the descriptive statistics, we described the data using the frequency distribution with mean or median values. Following this, measurement validation was conducted followed by a series of Ordinary Least Square (OLS) and logistic regression analyses to dissect potential correlations between immunisation completion status and the factors examined in this study (i.e., demographic, socio-economic, clinic-related, and parents-related variables). The software also provides a degree of correlation and variation between various independent and dependent variables [39].

## 5. Measurements

The childhood immunisation completion status was coded 0 if not completed and 1 if completed. Regarding the demographic variables, the gender of the parent and child was coded as 0 for male and 1 for female. The child’s age and birth order were treated as continuous variables. Regarding the socioeconomic variables, place of residence was coded as 0 for urban and 1 for rural, education was coded as 0 for less than university and 1 for university or higher, and accommodation type was coded as 0 if the accommodation is owned and 1 if the accommodation is rented. Employment status and housing type were coded as categorical variables. Employment status was coded as 0 for student, not employed, or other; 1 for government employee; and 2 for private sector employee. Housing type was coded as 0 for villa, 1 for flat, and 2 for mud house. The other socioeconomic variables (i.e., number of children, family income, and number of people living in the household) were treated as continuous variables. Regarding the clinic-related variables, the reminder system was coded as 0 if not available and 1 if available. Transportation type was coded as a categorical variable, with 0 for driving, 1 for walking, and 2 for walking and driving. Distance to PHC centres (in km), waiting time at the immunisation clinic, and clinic rating were coded as continuous variables. To evaluate the construct validity and reliability of the multi-item scales, exploratory factor analysis (EFA) was applied. As shown in Table 4, the items corresponding to each construct loaded well on separate factors.

More specifically, the EFA results demonstrate support for convergence and discriminant validity, as the constructs loaded well on distinct factors with minimal overlap with other constructs. For example, six items measuring attitude toward immunisation all loaded strongly on one factor and had negligible correlations with the other constructs. In other words, the items converge in measuring attitude toward immunisation and also diverge from other measurements (i.e., perceived risks, perceived benefits, knowledge, perceived behavioural control, and subjective social norms). In addition, Cronbach’s α for all multi-item constructs are all above the 0.70 threshold, supporting the reliability of the multi-item measurement scales for the parents-related factors. In summary, these analyses provide evidence that the scale used to measure the constructs are valid and reliable. Therefore, a mean score was computed for each construct (i.e., perceived risks, perceived benefits, knowledge, perceived behavioural control, and subjective social norms). The descriptive statistics of these constructs are provided in Table 4. 

## 6. Results

### 6.1. Correlation Analysis

As a preliminary analysis, we conducted a correlation test to assess the association between the completion status of childhood immunisation and all the other variables (i.e., demographic, socioeconomic, clinic-related, and parents-related variables), except the categorical variables. Table 5 presents the correlation matrix, which also shows the associations among all variables. Our interest in this preliminary analysis is to present an initial assessment of the associations involved in our theoretical framework. As seen in Table 5, nine variables (socioeconomic = 1, clinic-related = 3, and parents-related = 5) appear to correlate significantly with the completion status of childhood immunisation.

For example, accommodation type is negatively associated with the completion status of childhood immunisation (corr = −0.125, *p* = 0.019). This means that families in rented accommodation are less likely to complete childhood immunisation than those who own their accommodation. **Clinic-related variables**, such as distance to the PHC centre and clinic rating are positively associated with the completion status of childhood immunisation (corr = 0.200, *p* = 0.000; corr = 0.276, *p* = 0.000). While the former correlation is consistent with theory, the latter suggests that the longer the distance to the PHC centre, the more likely that families complete the childhood immunisation. However, this correlation does not hold, up as indicated by the regression analyses in which we control for all other variables to test the association between distance to the PHC centre and the completion status of childhood immunisation. Waiting time is negatively associated with the completion status of childhood immunisation (corr = −0.371, *p* = 0.000), suggesting that the longer parents wait, the less likely they are to complete their childhood immunisation. 

**Parents-related variables**, attitude, social norms, and perceived benefits are associated positively with the completion status of childhood immunisation (corr = 0.333, *p* = 0.000; corr = 0.356, *p* = 0.000; corr = 0.300, *p* = 0.000). In contrast, knowledge and perceived risks are associated negatively with the completion status of childhood immunisation (corr = −0.143, *p* = 0.007; corr = −0.245, *p* = 0.000). Behavioural control does not appear to correlate with the completion status of childhood immunisation (corr = −0.095, *p* = 0.076). While these correlations are useful to assess the associations we are interested in testing, a more robust approach to testing the associations is to control for all variables in one model. Next, we present the findings of a series of multiple regression analyses.

### 6.2. Hypothesis Testing and Regression Analysis

Table 6 presents the regression analysis results of demographic, socioeconomic, clinic-related, and parents-related variables. We first used OLS multiple regression. This is also referred to as a Linear Probability Model (LPM), given that the dependent variable (i.e., completion status of childhood immunisation) is binary. While this statistical approach is acceptable, it has some limitations. For example, the estimated coefficients can go outside the range of 0 and 1 in LPM [40]. To address this limitation, we also conducted the same analyses using logistic regression, which is a more appropriate statistical test in our case. As can be seen in Table 5, the results from the OLS and logistic regression are very similar, suggesting that our model is consistent and that the results do not depend on the estimation method. To start the analysis, we first regressed the demographic variables (Model 1), followed by the socioeconomic variables (Model 2), clinic-related variables (Model 3), and then parents-related variables (Model 4). Further, we conducted the last model (i.e., Model 5), in which we removed the variables that were not significantly associated with the completion status of childhood immunisation. We used the F test for joint significance, suggesting that the excluded variables together are not significantly associated with the completion status of childhood immunisation (F (16, 319) = 0.69, *p* = 0.80). Therefore, they were dropped from Model 5 which has also improved the model fit in the final model. Accordingly, Model 5 was relied upon to test our hypotheses.

H1 tests the association between demographic variables and the completion status of childhood immunisation. The results show that the age of the responding parent was negatively associated with the completion status of childhood immunisation, while the age of the other parent and the child’s age were positively associated with the completion status of childhood immunisation. This means that the older the responding parent, the less likely it is for the family to complete the childhood immunisation. In contrast, the older the other parent and the child’s age, the more likely the family will complete the childhood immunisation. These results are supported by both OLS and logistic regression.

Regarding H2, only two of the socioeconomic variables (father employment and housing type) were associated with the completion status of childhood immunisation. The findings suggest that fathers who work in the private sector are more likely to complete childhood immunisations than those who are unemployed or students. This finding is consistent across the OLS and logistic regression. Housing type is significantly associated with the completion status of childhood immunisation, suggesting that families living in a flat are less likely to complete the childhood immunisation. However, this result was not supported by the logistic model.

Regarding H3, transportation type and waiting time are both associated with the completion status of childhood immunisation. Specifically, families who drive to the PHC centre are more likely to complete the childhood immunisation as compared to those who walk or rely on both walking and driving. The waiting time is negatively associated with the completion status of childhood immunisation. This indicates that the more time parents wait to receive the childhood immunisation services, the less likely they will complete the immunisation for their children.

Parent-related hypotheses were aimed at testing the association between attitude (H4), social norms (H5), behavioural control (H6), knowledge (H7), perceived risks (H8), and perceived benefits (H9) and the completion status of childhood immunisation. Based on both the OLS and logistic regression, the results show that attitude, subjective social norms, and perceived benefits are positively associated with the completion status of childhood immunisation. Neither behavioural control nor perceived risks were associated with the completion status of childhood immunisation. Unexpectedly, the knowledge people have about immunisation is negatively and significantly associated with the completion status of childhood immunisation. This suggests that the more knowledge parents have (regardless of the truth of that knowledge), the less likely they will complete immunisation for their children. Interestingly, this negative association is the strongest among all other parents-related variables (i.e., attitude, social norms, behavioural control, perceived risks, and perceived benefits). This is evidenced by the logistic model coefficient (β = −0.714, s.e. = 0.134, *p* < 0.000), which is larger than other parents-related variables (see Table 6, Logistic Regression, Model 5). Table 7 provides a summary of the study’s hypothesis testing.

## 7. Discussion

This study identified several factors that have a significant impact on completion of childhood immunisation status of children in Saudi Arabia. These include, parental age, the employment type of father, the age of the child, transportation type, and waiting time at the PHC centre, as well as parents’ knowledge attitude, social norms, and perceived benefits.

Parental age was found to be negatively associated with the likelihood of immunising the child. This finding is generally consistent with previous reports of reduced knowledge and attitudes toward immunisation with the older age of parents [41,42]. Several potential factors can account for these findings. The parent-physician relationship plays a critical role in parents’ knowledge, attitude, and practise toward the immunisation of their children [43]. It is reported that younger parents have a more robust relationship with physicians in the context of the knowledge and safety profile of immunisation compared to older parents [42]. Parental age also plays a role in immunisation compliance towards children. Together, these factors may at least partly explain the negative association between parental age and child immunisation. 

In contrast to parental age, the age of the child was positively associated with the likelihood of immunisation. That is, older children were more likely to have completed their immunisation than younger children [44]. This is most likely related to the extended period allowing the completion, of the immunisation course. However, this does not indicate if immunisations were completed on time as there could be some delays.

Parents with private employment were more likely to immunise their children than unemployed or studying parents [45]. Private employment brings financial stability and most likely requires at least higher education, which could increase the likelihood of child immunisation. Conversely, unemployed or studying parents may not have adequate financial resources or education to vaccinate their children [46]. These findings are consistent with previous reports indicating a direct association of parents’ employment type with the immunisation status of their children [47,48,49]. We also found a relatively moderate significant association between residence type and immunisation status. Thus, families living in apartments were less likely to complete their children’s immunisation as compared to those who live in a villa. This is probably related to the financial situation since families with relatively moderate incomes tend to live in apartments rather than in large houses or villas [46]. In support of this, the association between parental income and childhood immunisation is well recognised.

These findings are further strengthened by the association between the mode of transportation and childhood immunisation. Wealthy parents are more likely to drive to PHC centres as compared to parents with moderate to low level income. Consistent with this, parents who drove their children to PHC were more likely to complete the childhood immunisation than parents who walked with or without driving. Upon reaching PHC centres, parents had to wait for their children’s immunisation. As expected, a longer waiting time at PHC centres was negatively associated with reduced chances of childhood immunisation. This finding is consistent with the literature, as the under-immunisation of children is partly due to negative experiences of parents at the health clinic, including a longer waiting time [50]. This finding emphasises the importance of optimising PHC services to reduce the waiting time and improve the prevalence of childhood immunisation.

The associations between parents’ attitudes, behavioural norms, and perceived benefits of childhood immunisation were also investigated. In general, these factors exhibited robust associations with the immunisation status of children. These findings align with our hypothesis and previous literature, as parents with positive attitudes and behavioural norms, as well as an adequate perception of the benefits of immunisation, are more likely to vaccinate their children [41,42]. The parental perception of immunisation is a critical driver for childhood immunisation since parents with a negative perception of immunisation are unlikely to vaccinate their children. This is evident by several studies indicating a robust association between negative parental perception of vaccines and reduced likelihood to immunise their children. Our data validates and extends these reports to include multiple regions of Saudi Arabia.

Interestingly, we found a negative association between immunisation knowledge and childhood immunisation completion. However, the survey did not investigate the scientific authenticity of the knowledge. This finding most likely indicates that parents’ knowledge about immunisation was scientifically inaccurate and primarily based on vaccine safety and efficacy misperceptions. This finding is supported by a previous report indicating that parental misconceptions about immunisations result in reduced immunisation rates among their children [51]. Thus, in order to improve childhood immunisation, it is essential to provide parents with accurate and relevant information regarding the safety and efficacy of vaccines.

One of the strengths of this study is the inclusion of several geographical regions of Saudi Arabia within the Qatif area, which minimises the potential confounding effects of regional cultures, ethnicities, and diverse socioeconomics on parental knowledge, attitude, and practise of childhood immunisation. The multicentric design of the study improves the generalisability of our findings and ensures potential comparisons among different centres.

Other studies conducted in Saudi Arabia including [8,11,18,19] have described several factors parents believe that they impact the national childhood immunisation compliance in the kingdom. The limitation with these studies is that they have been limited to parents’ experiences from urban areas of Saudi Arabia. We believed that a more comprehensive examination of these factors was required in both city and rural dwelling families. The major contribution of our study is that we added further investigations to what has been done in the past. Our investigation also included a determination of whether these studied characteristics (e.g., sociodemographic characteristics) differ in more rural communities. Another contribution is that we reviewed relevant literature and tested a number of hypotheses related to the association between demographic, socio-economic, clinic-related, and parents-related variables and childhood immunisation completion rate. To the best of our knowledge, this is the only study of its kind being conducted in Saudi Arabia and particularly in the Eastern Province. Therefore, this study contributes to the identification of critical factors associated with completion of childhood immunisation and presents important implications to healthcare practitioners, particularly in Saudi Arabia.

The present study has some limitations. The cross-sectional design of the study and non-probability sampling method limit any causal inferences and the generalisability of the results to the Saudi population. However, the current study is the first empirical investigation of the childhood immunisation in the Eastern Province of Saudi Arabia, the largest province in the kingdom. In addition, a potential recall bias may exist about undocumented data, such as the length of waiting time at PHC centres. It is also possible that some survey questions are under or over-estimated by parents, introducing a non-differential bias. Our data only includes parents attending PHC centres, and this selection bias should be considered during data interpretation. Thus, it is possible that the parents who refused immunisation did not attend the PHC centre and were excluded from this study. The respondents were likely to answer more favourably, introducing a potential social desirability bias. These limitations open avenues for future research and directions aimed at enhancing childhood immunisation, particularly in Saudi Arabia. For instance, future research may examine our model by PHC which may reveal variances across PHC that we could not detect in the current study as our sample would be too small for subgroup analyses. More specifically, future research needs to focus on comparisons between PHCs, place of residence, and other factors and tests whether the size of the detected effects is influenced by these factors.

## 8. Conclusions

The present study emphasises the importance of demographic, socio-economic, clinic-related, and parents-related variables that are associated with children immunisation. Several factors pertinent to Saudi Arabian parents may affect their decisions to vaccinate their children. Among them, several demographic and socioeconomic variables, proximity to a PHC centre, as well as the knowledge, attitude, and perception of immunisation of parents emerged as critical drivers of childhood immunisation. Our findings are clinically relevant in formulating strategies to improve childhood immunisation. Future studies are necessary to rigorously characterise the factors driving parental knowledge, attitude, and perception of childhood immunisation.

## Figures and Tables

**Figure 1 vaccines-10-02147-f001:**
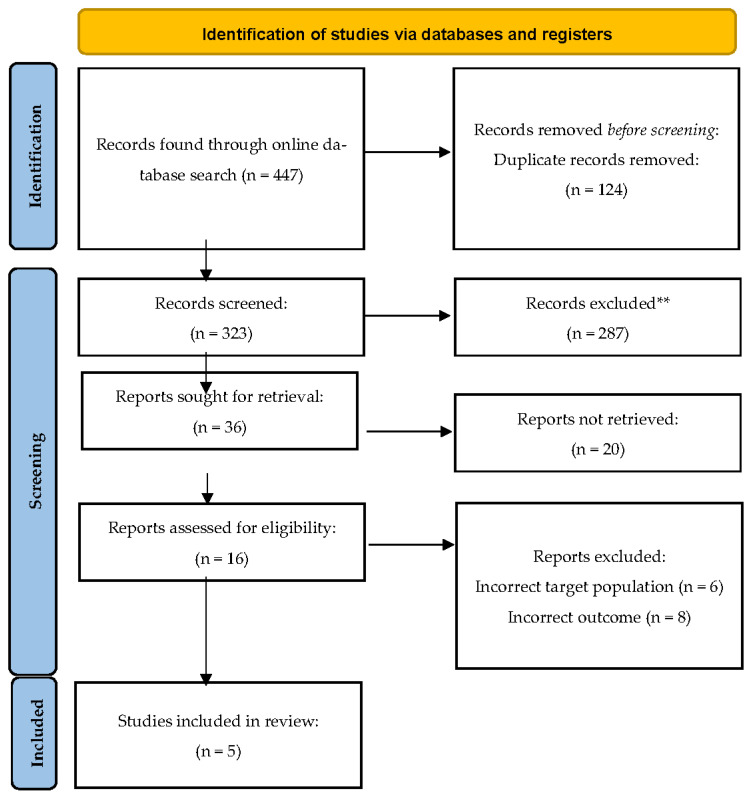
Flow diagram of the literature review.

**Figure 2 vaccines-10-02147-f002:**
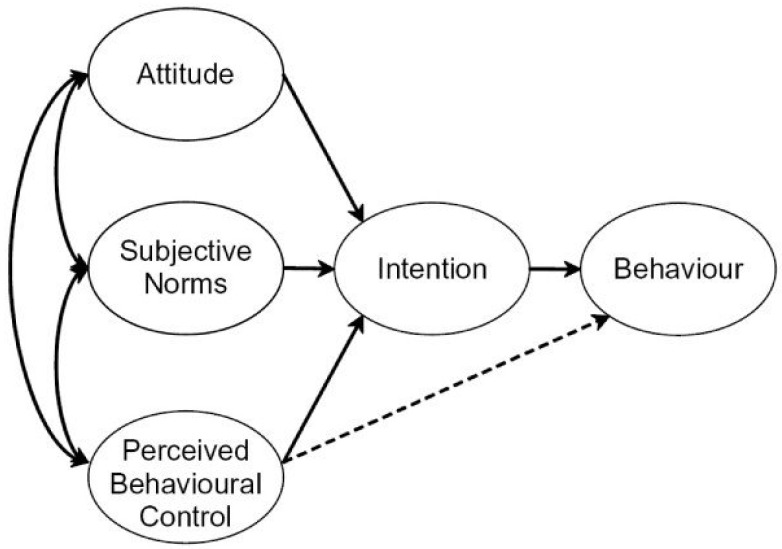
Constructs of the Theory of Planned Behaviour [34].

**Figure 3 vaccines-10-02147-f003:**
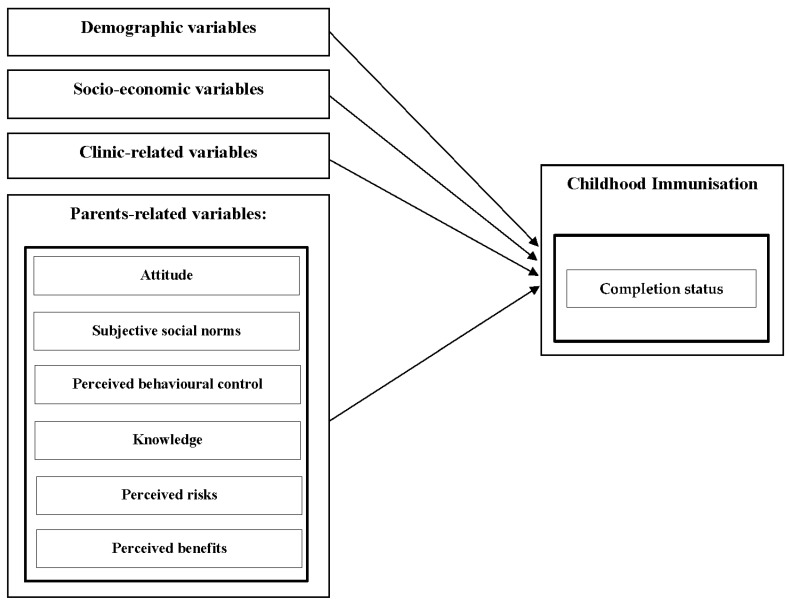
Framework for the study.

**Table 2 vaccines-10-02147-t002:** Descriptive statistics of all variables.

Completion Status of Childhood Immunisation	no = 29.70% yes = 70.30%	Have you completed the required immunisation for your child presented in the clinic today as of now (as per the immunisation schedule)?
**Demographic Variables**		
Gender	female = 55.00%; male = 45.00%	What is your gender (parent filling this questionnaire)?
Age	*mean* = 33.65; *s.d.* = 7.04; *min* = 19; *max* = 60	What is your age (parent filling this questionnaire)?
Age (other parent)	*mean* = 33.80; *s.d.* = 7.19; *min* = 19; *max* = 56	What is the other parent’s age?
Child Age (in months)	*mean* = 16.33; *s.d.* = 11.75; *min* = 1; *max* = 96	What is the age of your child?
Child Gender	female = 45.60%	What is the gender of your child?
Child Order	1st–2nd = 51.80% 3rd–4th = 37.40% 5th–6th = 7.90% 7th–8th = 2.00% 9th and over = 0.80%	What’s the order of your child in the family?
**Socio-economic Variables**		
Number of Children	*mean* = 2.72; *s.d.* = 1.54; *min* = 0; *max* = 9	What is the number of children in the family?
Place of Residence	urban = 43.30 rural = 56.70%	What is the place of your residence?
Father Education	less than university = 14.40% university or higher = 85.60%	What is the highest level of education (father of child)?
Mother Education	less than university = 28.00% university or higher = 72.00%	What is the highest level of education (mother of child)?
Father Employment	student, not employed, or other = 7.10% government employee = 34.00% private sector employee = 58.90%	What is the employment status (father of child)?
Mother Employment	student, not employed, or other = 50.40% government employee = 31.70% private sector employee = 17.80%	What is the employment status (mother of child)?
Family Income	Less than 5000 SAR = 7.90% 5001–10,000 SAR = 26.90% 10,001–15, 000 SAR = 25.50% 15,001–20,000 SAR = 16.70% More than 20,000 SAR = 22.90%	What is the family of child monthly income?
Number of People Living in Household	*mean* = 4.95; *s.d.* = 2.31; *min* = 1; *max* = 20	How many do people live in the household?
Housing Type	villa = 31.70% flat = 65.70% mud house or alike = 2.50%	Select the style of housing that the child and family live at.
Accommodation Type	owned = 64.30% rent = 35.70%	What is the accommodation type?
**Clinic-related Variables**		
Distance to PHC Centre (in km)	*mean* = 2.50; *s.d.* = 1.41; *min* = 1; *max* = 6	Distance to PHC centre (in kilometre).
Transportation Type	car = 69.40% walk = 12.70% car and walk = 17.80%	Mode of transport (traveling type).
Waiting Time	Less than 15 min = 15.60% 15–30 min = 42.50% 30–45 min = 19.00% 45–60 min = 12.50% More than 60 min = 10.50%	Waiting time at the immunisation clinic.
Clinic Rating	*mean* = 2.52; *s.d.* = 1.00; *min* = 1; *max* = 4	How do you rate the immunisation clinic facilities? (poor; fair; good; excellent)
Reminder System	not available = 37.70% available = 62.30%	Immunisation Reminder System (Does your clinic send you a reminder when your child is due for their next immunisation?)
**Parents-related Variables**		
Attitude	*mean* = 5.73; *s.d.* = 1.56; *min* = 1; *max* = 7	6 items; 7-point semantic scale
Social Norms	*mean* = 5.95; *s.d.* = 1.20; *min* = 1; *max* = 7	3 items; 7-point Likert scale
Behavioural Control	*mean* = 4.96; *s.d.* = 1.97; *min* = 1; *max* = 7	3 items; 7-point Likert scale
Knowledge	*mean* = 4.80; *s.d.* = 1.70; *min* = 1; *max* = 7	4 items; 7-point Likert scale
Perceived Risks	*mean* = 3.05; *s.d.* = 2.05; *min* = 1; *max* = 7	4 items; 7-point Likert scale
Perceived Benefits	*mean* = 5.78; *s.d.* = 1.49; *min* = 1; *max* = 7	4 items; 7-point Likert scale

**Table 3 vaccines-10-02147-t003:** Primary Health Care centres with identified study participation rate.

PHC	Frequency	Percent
Saihat 1	11	3.1%
Saihat 2	12	3.4%
Albustan	4	1.1%
Qudaih	12	3.4%
Aum Alhamam	11	3.1%
Alnasrah	21	5.9%
Alnabiah	2	0.6%
Albuhari	9	2.5%
Qatif 3	32	9.1%
Shuwaikha	8	2.3%
Dareen	13	3.7%
Rabiaa	6	1.7%
Sanabis	15	4.2%
Tarut	32	9.1%
Mahdood	11	3.1%
Auwammiah	26	7.4%
Tubi	6	1.7%
Khuwildiah	7	2.0%
Hilah	11	3.1%
Malahah	5	1.4%
Jish	13	3.7%
Muneera	20	5.7%
Majidiah	17	4.8%
Qudaih 2	9	2.5%
Ridah	5	1.4%
Saihat 3	7	2.0%
Reef	1	0.3%
Aljaroodia	27	7.6%
Total	353	100.00%

**Table 4 vaccines-10-02147-t004:** Exploratory Factor Analysis.

		Factor					
		α = 0.967	α = 0.966	α = 0.945	α = 0.829	α = 0.887	α = 0.878
Attitude 1	(bad … good)	**0.824**	−0.162	0.232	0.066	0.065	0.177
Attitude 2	(foolish … wise)	**0.823**	−0.174	0.255	0.022	0.072	0.113
Attitude 3	(unfavourable … favourable)	**0.793**	−0.207	0.266	0.002	0.052	0.204
Attitude 4	(useless … useful)	**0.843**	−0.235	0.218	−0.005	0.040	0.117
Attitude 5	(detrimental … beneficial)	**0.875**	−0.204	0.185	−0.043	0.062	0.130
Attitude 6	(unhealthy … healthy)	**0.875**	−0.194	0.179	−0.005	0.052	0.161
Perceived Risks 1	I am worried that immunisations might not be safe.	−0.255	**0.878**	−0.188	−0.010	0.037	−0.053
Perceived Risks 2	I am worried that immunisations might have serious side effects.	−0.237	**0.930**	−0.166	−0.037	−0.018	−0.115
Perceived Risks 3	I am worried that immunisations might have serious future risks.	−0.203	**0.897**	−0.177	−0.039	−0.013	−0.136
Perceived Risks 4	I am worried that immunisations might not prevent infectious diseases.	−0.222	**0.827**	−0.174	−0.001	0.044	−0.107
Perceived Benefits 1	Immunisations have a positive impact on public health.	0.286	−0.207	**0.663**	0.155	−0.036	0.113
Perceived Benefits 2	Immunisations are important for the prevention of infectious diseases that can have very serious effects.	0.372	−0.236	**0.777**	0.171	−0.025	0.232
Perceived Benefits 3	Immunisations are important to protect the health of our community.	0.339	−0.228	**0.836**	0.154	0.012	0.221
Perceived Benefits 4	I would feel safe if my child gets vaccinated.	0.349	−0.266	**0.781**	0.101	0.041	0.263
Knowledge 1	I have good knowledge about immunisations.	−0.085	−0.023	0.012	**0.913**	−0.013	0.003
Knowledge 2	I know which immunisations my child needs.	−0.073	−0.035	0.062	**0.927**	0.003	0.022
Knowledge 3	Unvaccinated children are more resistant to infections diseases.	0.153	−0.007	0.321	**0.477**	−0.036	0.247
Knowledge 5	Serious side effects of immunisations are very rare.	0.100	−0.003	0.246	**0.543**	0.000	0.265
Perceived Behavioural Control 1	It is completely up to me whether or not I get my children vaccinated.	0.096	0.010	0.009	−0.026	**0.880**	−0.056
Perceived Behavioural Control 2	If I wanted to, I could get my children vaccinated.	0.031	0.027	−0.009	0.032	**0.835**	0.010
Perceived Behavioural Control 3	It is completely up to me if I want to get my children vaccinated.	0.067	−0.001	−0.014	−0.037	**0.833**	−0.057
Subjective Social Norms 1	In my community, most parents like me have their children vaccinated with all the recommended vaccines.	0.293	−0.125	0.254	0.195	−0.136	**0.626**
Subjective Social Norms 2	Most parents like me think that it’s important to get their children vaccinated.	0.382	−0.264	0.410	0.141	0.021	**0.650**
Subjective Social Norms 3	Most people who are important to me (family, friends) think that I should give my children the required immunisations as indicated in the national immunisation card.	0.351	−0.229	0.343	0.196	−0.070	**0.630**

Extraction Method: Maximum Likelihood. Rotation Method: Varimax with Kaiser Normalization.

**Table 5 vaccines-10-02147-t005:** Correlation Matrix.

**Correlation Matrix**	1	2	3	4	5	6	7	8	9	10	11	12	13	14	15	16	17	18	19	20	21	22	23	24
1-Completion Status of Childhood Immunisation	1	−0.082	−0.008	−0.028	0.058	−0.052	0.025	0.023	−0.069	−0.040	0.065	0.081	0.078	−0.125 *	0.200 **	−0.371 **	0.276 **	0.041	0.333 **	0.356 **	−0.095	−0.143 **	−0.245 **	0.300 **
2-Gender (female)	−0.082	1	−0.293 **	0.267 **	−0.094	0.028	−0.036	−0.015	−0.056	−0.099	−0.031	−0.248 **	−0.042	0.139 **	−0.100	−0.083	0.037	0.127 *	−0.160 **	−0.056	−0.012	0.131 *	0.111 *	−0.034
3-Age	−0.008	−0.293 **	1	0.729 **	0.122 *	0.012	0.598 **	0.620 **	0.181 **	−0.081	0.031	0.310 **	0.429**	−0.120 *	0.120 *	0.033	0.002	−0.159 **	−0.029	0.065	−0.096	0.177 **	0.069	0.014
4-Age (other parent)	−0.028	0.267 **	0.729 **	1	0.063	0.043	0.556 **	0.610 **	0.199 **	−0.158 **	0.049	0.147 **	0.404**	−0.035	0.058	0.032	−0.034	−0.107 *	−0.120 *	0.043	−0.106 *	0.254 **	0.154 **	−0.021
5-Child Age (in months)	0.058	−0.094	0.122 *	0.063	1	−0.007	0.051	0.115 *	0.052	0.088	0.014	0.036	0.127*	−0.003	0.026	0.135 *	−0.019	0.019	0.023	−0.042	0.154 **	0.008	−0.003	−0.033
6-Child Gender (female)	−0.052	0.028	0.012	0.043	−0.007	1	0.089	0.027	0.084	0.020	0.009	−0.017	−0.050	−0.012	−0.041	−0.073	0.063	−0.016	0.000	−0.013	−0.012	0.097	0.043	0.005
7-Child Order	0.025	−0.036	0.598 **	0.556 **	0.051	0.089	1	0.788 **	0.203 **	−0.123 *	−0.075	0.215 **	0.540**	−0.028	0.045	−0.014	0.038	−0.084	0.011	0.055	−0.032	0.122 *	0.090	0.059
8-Number of Children	0.023	−0.015	0.620 **	0.610 **	0.115 *	0.027	0.788 **	1	0.197 **	−0.151 **	−0.130 *	0.112 *	0.686**	−0.028	0.019	−0.035	0.085	−0.072	−0.014	0.047	−0.046	0.096	0.150 **	0.044
9-Place of Residence (rural)	−0.069	−0.056	0.181 **	0.199 **	0.052	0.084	0.203 **	0.197 **	1	−0.181 **	−0.108 *	−0.184 **	0.184**	0.050	−0.110 *	0.026	0.010	−0.126 *	−0.086	0.086	−0.079	0.157 **	0.034	0.052
10-Father Education (university or higher)	−0.040	−0.099	−0.081	−0.158 **	0.088	0.020	−0.123 *	−0.151 **	−0.181 **	1	0.338 **	0.326 **	−0.141**	0.003	0.062	0.178 **	−0.109 *	−0.022	0.036	−0.072	0.139 **	−0.049	−0.007	0.004
11-Mother Education (university or higher)	0.065	−0.031	0.031	0.049	0.014	0.009	−0.075	−0.130 *	−0.108 *	0.338 **	1	0.413 **	−0.100	−0.028	0.135 *	0.023	−0.096	−0.120 *	−0.087	−0.068	−0.027	−0.061	0.076	−0.080
12-Family Income	0.081	−0.248 **	0.310 **	0.147 **	0.036	−0.017	0.215 **	0.112 *	−0.184 **	0.326 **	0.413 **	1	0.044	−0.138 **	0.208 **	0.091	−0.118 *	−0.072	0.080	0.023	0.037	−0.116 *	−0.011	0.030
13-Number of People Living in Household	0.078	−0.042	0.429 **	0.404 **	0.127 *	−0.050	0.540 **	0.686 **	0.184 **	−0.141 **	−0.100	0.044	1	−0.096	0.056	−0.052	0.124 *	−0.120 *	0.034	0.069	−0.111*	0.110*	0.094	0.094
14-Accommodation Type (rent)	**−0.125 ***	0.139 **	−0.120 *	−0.035	−0.003	−0.012	−0.028	−0.028	0.050	0.003	−0.028	−0.138 **	−0.096	1	−0.122 *	0.047	−0.058	0.004	−0.071	−0.104	−0.037	−0.056	0.037	−0.080
15-Distance to PHC Centre (in km)	**0.200 ****	−0.100	0.120 *	0.058	0.026	−0.041	0.045	0.019	−0.110 *	0.062	0.135 *	0.208 **	0.056	−0.122 *	1	−0.042	0.003	−0.059	0.018	−0.005	−0.114 *	−0.152 **	−0.068	0.054
16-Waiting Time	**−0.371 ****	−0.083	0.033	0.032	0.135 *	−0.073	−0.014	−0.035	0.026	0.178 **	0.023	0.091	−0.052	0.047	−0.042	1	−0.668 **	−0.043	0.076	−0.095	0.202 **	−0.083	0.011	−0.021
17-Clinic Rating	**0.276 ****	0.037	0.002	−0.034	−0.019	0.063	0.038	0.085	0.010	−0.109 *	−0.096	−0.118 *	0.124*	−0.058	0.003	−0.668 **	1	0.181 **	0.113 *	0.158 **	−0.048	0.184 **	−0.153 **	0.167 **
18-Reminder System (available)	0.041	0.127 *	−0.159 **	−0.107 *	0.019	−0.016	−0.084	−0.072	−0.126 *	−0.022	−0.120 *	−0.072	−0.120 *	0.004	−0.059	−0.043	0.181 **	1	0.036	−0.018	0.138 **	−0.104	−0.048	−0.031
19-Attitude	**0.333 ****	−0.160 **	−0.029	−0.120 *	0.023	0.000	0.011	−0.014	−0.086	0.036	−0.087	0.080	0.034	−0.071	0.018	0.076	0.113 *	0.036	1	0.599 **	0.118 *	0.131 *	−0.476 **	0.619 **
20-Social Norms	**0.356 ****	−0.056	0.065	0.043	−0.042	−0.013	0.055	0.047	0.086	−0.072	−0.068	0.023	0.069	−0.104	−0.005	−0.095	0.158 **	−0.018	0.599 **	1	−0.068	0.385 **	−0.456 **	0.696 **
21-Behavioural Control	−0.095	−0.012	−0.096	−0.106 *	0.154 **	−0.012	−0.032	−0.046	−0.079	0.139 **	−0.027	0.037	−0.111 *	−0.037	−0.114 *	0.202 **	−0.048	0.138 **	0.118 *	−0.068	1	−0.031	0.009	0.011
22-Knowledge	**−0.143 ****	0.131 *	0.177 **	0.254 **	0.008	0.097	0.122 *	0.096	0.157 **	−0.049	−0.061	−0.116 *	0.110 *	−0.056	−0.152 **	−0.083	0.184 **	−0.104	0.131 *	0.385 **	−0.031	1	−0.104	0.365 **
23-Perceived risks	**−0.245 ****	0.111 *	0.069	0.154 **	−0.003	0.043	0.090	0.150 **	0.034	−0.007	0.076	−0.011	0.094	0.037	−0.068	0.011	−0.153 **	−0.048	−0.476 **	−0.456 **	0.009	−0.104	1	−0.499 **
24-Perceived Benefits	**0.300 ****	−0.034	0.014	−0.021	−0.033	0.005	0.059	0.044	0.052	0.004	−0.080	0.030	0.094	−0.080	0.054	−0.021	0.167 **	−0.031	0.619 **	0.696 **	0.011	0.365 **	−0.499 **	1

* *p* < 0.05; ** *p* < 0.01.

**Table 6 vaccines-10-02147-t006:** OLS and Logistic Regression Results.

	OLS Regression					Logistic Regression				
	Model 1	Model 2	Model 3	Model 4	Model 5	Model 1	Model 2	Model 3	Model 4	Model 5
Dependent Variable: Completion Status of Childhood Immunisation	*β*	*β*	*β*	*β*	*β*	*β*	*β*	*β*	*β*	*β*
	(*robust s.e.*)	(*robust s.e.*)	(*robust s.e.*)	(*robust s.e.*)	(*robust s.e.*)	(*robust s.e.*)	(*robust s.e.*)	(*robust s.e.*)	(*robust s.e.*)	(*robust s.e.*)
**Demographic Variables**										
Gender (female)	−0.135 *	−0.101	**−0.163 ****	**−0.0996 ***	-	−0.657 *	−0.660	−1.160 **	−0.781	-
	(0.0752)	(0.0781)	(0.0646)	(0.0552)	-	(0.368)	(0.424)	(0.451)	(0.503)	-
Age	−0.0109	−0.0126	**−0.0205 *****	**−0.0147 ****	**−0.00886 ****	−0.0528	−0.0846 *	**−0.158*****	**−0.161 *****	**−0.0975 ****
	(0.00817)	(0.00811)	(0.00655)	(0.00598)	(0.00418)	(0.0389)	(0.0454)	(0.0492)	(0.0616)	(0.0387)
Age (other parent)	0.00590	0.00461	**0.0137 ****	**0.0165 *****	**0.00982 ****	0.0283	0.0345	**0.101 ****	**0.166 *****	**0.100 *****
	(0.00746)	(0.00756)	(0.00639)	(0.00559)	(0.00406)	(0.0356)	(0.0408)	(0.0439)	(0.0527)	(0.0349)
Child Age (in months)	0.00214	0.00223	**0.00359 ***	**0.00459 ****	**0.00425 ****	0.0111	0.0134	0.0297	**0.0474 ***	**0.0440 ***
	(0.00186)	(0.00204)	(0.00206)	(0.00192)	(0.00182)	(0.0104)	(0.0122)	(0.0186)	(0.0243)	(0.0238)
Child Gender (female)	−0.0508	−0.0499	−0.0702	−0.0485	-	−0.245	−0.271	−0.510 *	−0.709 *	-
	(0.0498)	(0.0493)	(0.0444)	(0.0393)	-	(0.239)	(0.259)	(0.299)	(0.362)	-
Child Order	0.0407	0.0318	0.0355	0.0318	-	0.204	0.143	0.217	0.253	-
	(0.0335)	(0.0440)	(0.0354)	(0.0315)	-	(0.174)	(0.234)	(0.227)	(0.291)	-
**Socio-economic Variables**										
Number of Children		−0.0117	−0.0105	−0.0174	-		−0.148	−0.124	−0.141	-
		(0.0258)	(0.0217)	(0.0202)	-		(0.147)	(0.150)	(0.221)	-
Place of Residence (rural)		−0.0203	−0.0203	−0.0202	-		−0.0902	−0.192	−0.270	-
		(0.0558)	(0.0518)	(0.0461)	-		(0.286)	(0.353)	(0.469)	-
Father Education (university or higher)		−0.140 *	−0.0633	−0.0245	-		−0.762 *	−0.461	0.0932	-
		(0.0750)	(0.0676)	(0.0629)	-		(0.414)	(0.461)	(0.623)	-
Mother Education (university or higher)		0.0592	−0.00391	0.0532	-		0.326	0.0212	0.706	-
		(0.0679)	(0.0577)	(0.0487)	-		(0.340)	(0.371)	(0.452)	-
Father Employment (government employee)		0.153	0.160	0.123	0.122		0.730	1.115*	1.291 *	1.094
		(0.110)	(0.102)	(0.0912)	(0.0857)		(0.495)	(0.594)	(0.756)	(0.695)
Father Employment (private sector employee)		0.225 **	**0.186 ***	**0.153 ***	**0.150 ***		1.120 **	**1.297 ****	**1.466 ****	**1.262 ***
		(0.106)	(0.0992)	(0.0865)	(0.0825)		(0.484)	(0.581)	(0.704)	(0.653)
Mother Employment (government employee)		−0.00711	−0.00460	−0.0295	-		−0.0685	−0.0269	−0.509	-
		(0.0706)	(0.0629)	(0.0536)	-		(0.363)	(0.410)	(0.504)	-
Mother Employment (private sector employee)		0.137 *	0.117*	0.0488	-		0.869 *	0.904 *	0.387	-
		(0.0734)	(0.0670)	(0.0578)	-		(0.471)	(0.526)	(0.636)	-
Family Income		0.00198	0.00868	−0.0178	-		0.00858	0.0297	−0.271	-
		(0.0247)	(0.0236)	(0.0195)	-		(0.131)	(0.151)	(0.180)	-
Number of People Living in Household		0.0201 **	0.00585	0.00349	-		0.204 **	0.117	0.140	-
		(0.00957)	(0.00789)	(0.00745)	-		(0.0828)	(0.0826)	(0.145)	-
Housing Type (flat)		−0.164 ***	**−0.114 ****	**−0.0810 ***	**−0.0811 ****		−1.046 ***	**−0.920 ****	**−0.910 ***	−0.612
		(0.0589)	(0.0545)	(0.0462)	(0.0383)		(0.343)	(0.413)	(0.533)	(0.394)
Housing Type (mud house or alike)		−0.171	0.120	0.0115	0.0109		−1.050	0.591	−0.240	0.231
		(0.158)	(0.135)	(0.121)	(0.113)		(0.756)	(0.839)	(1.180)	(1.337)
Accommodation Type (rent)		−0.0379	−0.00106	−0.00562	-		−0.170	0.00293	−0.0558	-
		(0.0596)	(0.0510)	(0.0440)	-		(0.279)	(0.316)	(0.408)	-
**Clinic-related Variables**										
Distance to PHC Centre (in km)			0.0123	−0.00482	-			0.0289	−0.115	-
			(0.0166)	(0.0144)	-			(0.132)	(0.157)	-
Transportation Type (walking)			**−0.364 *****	**−0.339 *****	**−0.351 *****			**−2.082 *****	**−2.870 *****	**−2.672 *****
			(0.0783)	(0.0653)	(0.0606)			(0.465)	(0.616)	(0.511)
Transportation Type (walking and driving)			**−0.181 *****	**−0.218 *****	**−0.215 *****			**−1.059 *****	**−1.752 *****	**−1.547 *****
			(0.0633)	(0.0591)	(0.0562)			(0.373)	(0.541)	(0.445)
Waiting Time			**−0.0984 *****	**−0.122 *****	**−0.123 *****			**−0.554 *****	**−0.990 *****	**−0.913 *****
			(0.0253)	(0.0222)	(0.0156)			(0.162)	(0.235)	(0.170)
Clinic Rating			0.0377	0.00446	-			0.280	0.0619	-
			(0.0288)	(0.0255)	-			(0.184)	(0.230)	-
Reminder System (available)			−0.00754	−0.0157	-			−0.157	−0.291	-
			(0.0477)	(0.0429)	-			(0.327)	(0.433)	-
**Parents-related Variables**										
Attitude				**0.0510 *****	**0.0507 *****				**0.371 *****	**0.337 *****
				(0.0182)	(0.0176)				(0.109)	(0.106)
Social Norms				**0.0836 *****	**0.0903 *****				**0.653 *****	**0.672 *****
				(0.0233)	(0.0219)				(0.197)	(0.185)
Behavioural Control				−0.00486	−				−0.0126	−
				(0.0104)	−				(0.0991)	−
Knowledge				**−0.0810 *****	**−0.0848 *****				**−0.726 *****	**−0.714 *****
				(0.0117)	(0.0102)				(0.162)	(0.134)
Perceived Risks				−0.00260	−				−0.0526	-
				(0.0121)	−				(0.109)	-
Perceived Benefits				**0.0486 ****	**0.0468 ****				**0.390 ****	**0.379 *****
				(0.0202)	(0.0191)				(0.159)	(0.132)
Constant	0.863 ***	0.895 ***	1.079 ***	0.352	0.232	1.647 **	2.303 **	3.765 ***	−0.525	−1.815
	(0.148)	(0.203)	(0.234)	(0.237)	(0.167)	(0.726)	(1.028)	(1.455)	(2.039)	(1.464)
Observations	351	351	351	351	351	351	351	351	351	351
*R-squared* (*Pseudo R-squared*)	0.019	0.107	0.314	0.509	0.494	(0.015)	(0.100)	(0.280)	(0.499)	(0.473)
*Adjusted R-squared*	0.001	0.055	0.260	0.460	0.473	-	-	-	-	-
*F-value* (*Wald Chi-squared*)	1.11	3.12 ***	10.03 ***	22.09 ***	38.81 ***	(6.14)	(41.55 ***)	(83.62 ***)	(104.66 ***)	(88.05 ***)

*** *p* < 0.01, ** *p* < 0.05, * *p* < 0.1.

**Table 7 vaccines-10-02147-t007:** Summary of hypothesis testing results.

Hypothesis	Support
**H1:** *Demographic variables (e.g., age, gender, etc.) are associated with childhood immunisation completion status.*	Supported for parental age, age of the other parent, and age of the child
**H2:** *Socioeconomic variables (e.g., place of residence, level of education, etc.) are associated with childhood immunisation completion status.*	Supported for father employment and housing type
**H3:** *Clinic-related variables (e.g., distance to PHC centre, waiting time at immunisation clinic, etc.) are associated with childhood immunisation completion status.*	Supported for transportation type and waiting time
**H4:** *A positive attitude toward immunisation is associated with childhood immunisation completion status.*	Supported
**H5:** *Subjective social norms toward immunisation are positively associated with childhood immunisation completion status.*	Supported
**H6:** *Perceived behavioural control toward immunisation is positively associated with childhood immunisation completion status.*	Not supported
**H7:** *Higher knowledge about immunisation is positively associated with childhood immunisation completion status.*	Not supported but significant in the opposite direction
**H8:** *Higher perceived risks of immunisation are negatively associated with childhood immunisation completion status.*	Not supported
**H9:** *Higher perceived benefits of immunisation are positively associated with childhood immunisation completion status.*	Supported

## Data Availability

The online questionnaires are hosted in LimeSurvey (https://www.limesurvey.org/privacy-policy (accessed on 7 December 2022)) supported by the University of Newcastle, Australia. Additionally, all data collected will be stored on the University of Newcastle’s Cloud secure server for a minimum of 5 years and will only be accessible to members of the research team. Data will be securely destroyed in line with UON policy provisions.

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
