# Peer review of "Exploring Critical Factors Associated with Completion of Childhood Immunisation in the Eastern Province of Saudi Arabia"

_vaccines, 2022, doi:10.3390/vaccines10122147_

Round 1
Reviewer 1 Report
The manuscript aimed to explore critical factors associated with completion of childhood immunisation to the National Childhood Immunisation Programme and the resulting immunisation compliance across the rural and urban areas of Qatif, located in the Eastern Province of Saudi Arabia. Overall, the argument of the manuscript is original and it is of interest for the scientific literature. The introduction and discussion sections lack contents on the implications this study could have on national policies and what the paper adds to scientific literature. It is necessary that the authors investigate these aspects in depth, highlighting the areas in which these results could have healthcare implications. In general, the English in the paper can be understood. It contains some language errors or bad constructed sentences so authors should check carefully the English.
I propose the following comments and suggestions for the authors to consider.
The Major Essential Revisions include:
1. Lines 42-44: authors should report a reference to justify the sentence
2. Authors should provide the precise reference to the ethical approval obtained from the University of Newcastle. Did you have a code number as reference?
3. The discussion section does not provide sufficient information and in-depth discussion. It does not contain new results that significantly advance the research field. Could you expand a bit more on how your study differed/added to literature?
Author Response
Response to reviewer 1:
Below please find our responses.
The Major Essential Revisions include:
- Lines 42-44: authors should report a reference to justify the sentence
Thank you very much for providing this comment. As suggested, three references have been added, and it reads much better now. The sentence reads as follows:
“Children are specifically vulnerable to acquiring infectious diseases due to their un-trained and undeveloped immune systems, highlighting the importance of immunising children against infectious diseases [3-5].”
- Authors should provide the precise reference to the ethical approval obtained from the University of Newcastle. Did you have a code number as reference?
We have two approvals: one from the University of Newcastle and the other one from the Saudi Ministry of Health. We agree with you that this is a very important point and adding the reference/code numbers obtained from both organizations will provide more credibility. Therefore, both reference numbers have been added to the paper under the recruitment and data collection section (section 4.3). The following statement added to the same section:
“This study obtained ethical approval from the University of Newcastle, Australia (ethics reference no. H-2021-0378), and the Ministry of Health of Saudi Arabia (ethics reference no. QCH-SREC07/2022).”
- The discussion section does not provide sufficient information and in-depth discussion. It does not contain new results that significantly advance the research field. Could you expand a bit more on how your study differed/added to literature?
Thank you for this constructive comment and we do understand that the previous submission did not provide sufficient information and in-depth discussion related to how our work advances the literature. In the revised version, we added a paragraph that discusses our contributions in more details. Here’s what has been added:
“Other studies conducted in Saudi Arabia including [11,18-20] have described several factors parents believe that they impact the national childhood immunisation compliance in the kingdom. The limitation with these studies is that they have been limited to parents’ experiences from urban areas of Saudi Arabia. We believed that a more comprehensive examination of these factors was required in both city and rural dwelling families. The major contribution of our study is that we added further investigations to what has been done in the past. Our investigation also included a determination of whether these studied characteristics (e.g., sociodemographic characteristics) differ in more rural communities. Another contribution is that we reviewed relevant literature and tested a number of hypotheses related to the association between demographic, socioeconomic, clinic-related, and parents-related variables and childhood immunisation completion rate. To the best of our knowledge, this is the only study of its kind being conducted in Saudi Arabia and particularly in the Eastern Province. Therefore, this study contributes to the identification of critical factors associated with completion of childhood immunisation and presents important implications to healthcare practitioners, particularly in Saudi Arabia.”
Reviewer 2 Report
Marwa Alabadi and colleagues conducted research on the factors influencing parents’ completion of childhood immunisation in the Eastern Province of Saudi Arabia. The results found that attitudes, social norms, awareness of immunisation, and parents working in private companies were more likely to vaccinate their children. The study is interesting and preliminarily identifies several factors that have a positive or negative impact on childhood immunization completion. However, there are some problems:
1. In the abstract, the content of "2. Literature Review" is not included;
2. Few keywords;
3. Only 5 studies were included in the literature studies, but all of them were cross-sectional observations, and the results were reliable;
4. The study was conducted in Qatif, located in Saudi Arabia's Eastern Province. Can it be considered representative of the Eastern Province of Saudi Arabia?
5. 351 samples are distributed in 27 PHCs, with an average of only 13 samples per PHC, is it too little?
6. Of the 27 PHCs, 20 in urban areas and 7 in rural areas, does the sampling take into account equilibrium?
7. What is the occupational distribution of fathers in rural areas, where 34 percent of fathers are government employees and 59 per cent are private sector employees, for a total of 93 percent?
8. The three structures that determine the outcome of behavior according to the TPB are attitudes, subjective norms, and perceptual behavior control, but the three structures described below are behavioral intentions, attitudes, and subjective norms. Why are they inconsistent?
9. Are there differences between urban and rural outcomes examined in subgroup analyses?
Author Response
Response to reviewer 2:
Below please find our responses.
- In the abstract, the content of "2. Literature Review" is not included;
- Thank you very much for providing this comment. We have updated the abstract accordingly. It now reads as follow “(2) Literature review: a systematic literature review was conducted to understand what is currently published concerning parents’ immunisation compliance in Saudi Arabia and the factors associated with immunisation compliance”
- Few keywords
- We initially had a few more keywords to include but we thought the journal required no more 3 keywords. Considering your comment, we have inserted a few more relevant keywords and they read as follow “public healthcare; childhood immunisation; childhood vaccination; vaccine hesitancy; delayed vaccination; Saudi Arabia”
- Only 5 studies were included in the literature studies, but all of them were cross-sectional observations, and the results were reliable;
- We completely agree with you as there’s lack of literature in this domain in Saudi Arabia. Most of the studies found in recent literature are descriptive with lack of investigation in depth. The studies were excluded based on the criteria which we provided in section 2.1.3. In summary, 447 studies were found in the first stage of the literature review. After removing duplicates, 323 studies were reviewed for relevance by reading the titles and abstracts. Of these, 16 were screened based on the inclusion and exclusion criteria, and eleven studies were excluded because of their incorrect target populations or outcomes. Five studies were identified as relevant due to their relevance and potential significance. Most importantly, we couldn’t find any studies covering the targeted population of the city of Qatif. It is worth mentioning that we previously published a more comprehensive systematic literature review paper in this journal and we have cited the work in the manuscript so the readers could go back to gain insights of what has been done in the past. Here’s the reference of our previous published paper:
- Alabadi, M.; Aldawood, Z. Parents’ Knowledge, Attitude and Perceptions on Childhood Vaccination in Saudi Arabia: A Systematic Literature Review. Vaccines 2020, 8, 750. https://doi.org/10.3390/vaccines8040750
- We completely agree with you as there’s lack of literature in this domain in Saudi Arabia. Most of the studies found in recent literature are descriptive with lack of investigation in depth. The studies were excluded based on the criteria which we provided in section 2.1.3. In summary, 447 studies were found in the first stage of the literature review. After removing duplicates, 323 studies were reviewed for relevance by reading the titles and abstracts. Of these, 16 were screened based on the inclusion and exclusion criteria, and eleven studies were excluded because of their incorrect target populations or outcomes. Five studies were identified as relevant due to their relevance and potential significance. Most importantly, we couldn’t find any studies covering the targeted population of the city of Qatif. It is worth mentioning that we previously published a more comprehensive systematic literature review paper in this journal and we have cited the work in the manuscript so the readers could go back to gain insights of what has been done in the past. Here’s the reference of our previous published paper:
- The study was conducted in Qatif, located in Saudi Arabia's Eastern Province. Can it be considered representative of the Eastern Province of Saudi Arabia?
-
- Given that our data collection was not based on a probability sampling method, we cannot claim that the results are representative. In the revised version, we indicate this in the limitation section and motivate future research especially that this is the first study conducted in Qatif. The revised paragraph reads as follow:
-
-
- “The present study has some limitations. The cross-sectional design of the study limits any causal inferences and the generalisability of the results to the Saudi population. However, the current study is the first empirical investigation of the childhood immunisation in the Eastern Province of Saudi Arabia, the largest province in the kingdom. In addition, a potential recall bias may exist about undocumented data, such as the length of waiting time at PHC centres. It is also possible that some survey questions are under or over-estimated by parents, introducing a non-differential bias. Our data only includes parents attending PHC centres, and this selection bias should be considered during data interpretation. Thus, it is possible that the parents who refused immunisation did not attend the PHC centre and were excluded from this study. The respondents were likely to answer more favourably, introducing a potential social desirability bias. These limitations open avenues for future research and directions aimed at enhancing childhood immunisation, particularly in Saudi Arabia. For instance, future research may examine our model by PHC which may reveal variances across PHC that we could not detect in the current study as our sample would be too small for subgroup analyses. More specifically, future research needs to focus on comparisons between PHCs, place of residence, and other factors and tests whether the size of the detected effects is influenced by these factors.”
-
- 351 samples are distributed in 27 PHCs, with an average of only 13 samples per PHC, is it too little?
-
- When considering the data in that way, indeed this is a small sample per PHC. This is why we do not control for this variable in our analyses as the many categories with small observations would require a much larger sample size than we currently have (n =351). We wish we could collect more data but the resources we had were limited. The sample size of 351, however, is sufficient to detect significant correlations among our variables of interest as we did not aim to analyse the data by PHCs. We hope this addresses your comment.
- Of the 27 PHCs, 20 in urban areas and 7 in rural areas, does the sampling take into account equilibrium?
-
- As mentioned in the previous comment, the unit of analysis in our study is the parent of the child regardless of the PHCs or the area they belong to. However, we control for these variables in the statistical models to partial out their effect when testing our hypotheses.
- What is the occupational distribution of fathers in rural areas, where 34 percent of fathers are government employees and 59 per cent are private sector employees, for a total of 93 percent?
-
- As shown in Table 2, 7% are indicated as student, not employed, or other and this would result in 100%.
- The three structures that determine the outcome of behavior according to the TPB are attitudes, subjective norms, and perceptual behavior control, but the three structures described below are behavioral intentions, attitudes, and subjective norms. Why are they inconsistent?
-
- Thank you for this insightful comment. We have revised the paragraph that describes TPB and addressed the noted inconsistency. “The first construct is attitude towards the behaviour, which is the extent to which a person has a favourable or unfavourable appraisal of a given behaviour. Attitudes consist of behavioural beliefs and outcome evaluations. The second construct is subjective norms, which reflect the social pressure to perform or refrain from performing a given behaviour. Subjective norms consist of normative beliefs and the motivation to comply. The third construct is perceived behavioural control which refers to people’s perception of the ease or difficulty of performing the behaviour of interest. According to TPB, these constructs determine behavioural intention, which is the motivational factor that influences actual behaviour [32]. The stronger the intention to engage in a given behaviour, the more likely it is to perform that behaviour.”
- Are there differences between urban and rural outcomes examined in subgroup analyses?
-
- This is outside the scope of our paper and as mentioned above our sample size would be very small for subgroup analyses. Therefore, we noted this in the limitation section and motive future research to examine subgroup analyses.
Reviewer 3 Report
First of all, I would like to thank for the opportunity to review this paper. Actually, the vaccination campaign is the first method to counteract infectious diseases; however, sufficient vaccination coverage is conditioned by the people’s acceptance of these vaccines. In this context, the paper under review is aimed at evaluate the association between demographic, socio-economic, clinic-related, and parents-related variables and completion of childhood immunisation.
The subject under study is certainly very important, as general issue in public health. The article presents interesting results but, but it is nevertheless believed that, given the organization of the contents and the description of the same, the manuscript cannot be published in its current form, especially for its local impact and the small sample. I would like to encourage authors to consider several issues to be improved.
Title: it must be improved. It should better highlight the object of the study.
Introduction: The authors should make clearer what is the gap in the literature that is filled with this study. The authors do not frame their study in the frame of 21th century emerging priorities that can influence the vaccine compliance (refer to articles with DOI: 10.3390/ijerph191911929), after COVID-19 many health related issues were changed. What is the possible international contribution of the study to the literature? The objectives should be better explained at the end of the section.
Methods: The survey was conducted using a non-standard questionnaire. The use of an unreliable instrument is a serious and irreversible limitation of the study. The fact that a similar questionnaire has been used in previous surveys is not sufficient. A validation process must be performed to evaluate the tool in a different population. What about face validity and intelligibility? There was a pilot study?
The enrolment procedure must be better specified. How did the authors choose the way to select the sample? This can represent a great bias origin. How did they avoid the selection bias?
Statistical analysis: I suggest to insert a measure of the magnitude of the effect for the comparisons. Please consider to include effect sizes.
Discussion: I also suggest expanding. Emphasize the contribution of the study to the literature in terms of public health.
Author Response
Response to reviewer 3:
Below please find our responses.
- Title: it must be improved. It should better highlight the object of the study.
- Thank you for your valuable comments. We have adjusted the title to match the aim of the study to the following: “Exploring Critical Factors Associated with Completion of Childhood Immunisation in the Eastern Province of Saudi Arabia.”
- Introduction: The authors should make clearer what is the gap in the literature that is filled with this study. The authors do not frame their study in the frame of 21th century emerging priorities that can influence the vaccine compliance (refer to articles with DOI: 10.3390/ijerph191911929), after COVID-19 many health related issues were changed. What is the possible international contribution of the study to the literature? The objectives should be better explained at the end of the section.
-
- Thank you for commenting on point. As a result, we added the suggested reference to the last paragraph of the introduction.
-
- This point covers the contribution
-
-
- In the revised version, we added a paragraph that discusses our contributions in more details. Here’s what has been added:
-
-
-
-
- “Other studies conducted in Saudi Arabia including [11,18-20] have described several factors parents believe that they impact the national childhood immunisation compliance in the kingdom. The limitation with these studies is that they have been limited to parents’ experiences from urban areas of Saudi Arabia. We believed that a more comprehensive examination of these factors was required in both city and rural dwelling families. The major contribution of our study is that we added further investigations to what has been done in the past. Our investigation also included a determination of whether these studied characteristics (e.g., sociodemographic characteristics) differ in more rural communities. Another contribution is that we reviewed relevant literature and tested a number of hypotheses related to the association between demographic, socioeconomic, clinic-related, and parents-related variables and childhood immunisation completion rate. To the best of our knowledge, this is the only study of its kind being conducted in Saudi Arabia and particularly in the Eastern Province. Therefore, this study contributes to the identification of critical factors associated with completion of childhood immunisation and presents important implications to healthcare practitioners, particularly in Saudi Arabia.”
-
-
- Methods: The survey was conducted using a non-standard questionnaire. The use of an unreliable instrument is a serious and irreversible limitation of the study. The fact that a similar questionnaire has been used in previous surveys is not sufficient. A validation process must be performed to evaluate the tool in a different population. What about face validity and intelligibility? There was a pilot study?
- We have conducted both construct validity and reliability tests to validate the instruments we used in our questionnaire. Please refer to Section 5 and specifically Table 4 for the empirical evidence. The face validity and intelligibility of the instrument were assessed with experts in the field before we launched the data collection. We also conducted a pilot study in which we collected 30 observations to test whether subjects had any comprehension issues with our questions and whether there are any unexpected readability issues. We did not detect any issues and hence we launched the data collection of the reported study.
- The enrolment procedure must be better specified. How did the authors choose the way to select the sample? This can represent a great bias origin. How did they avoid the selection bias?
-
- Thank you very much for this insightful point. However, we included the following statement in section 4.2 (study design and sample size determination) which explains how we selected the participants using a scientific method.
-
-
- “A statistical power analysis was conducted to determine the sample size. Based on Qatif’s population of 1,100,000 and the need to keep the confidence interval as 95%, the margin of error as 5%, and population proportion as 50%, we need around 350 parents visiting 27 selected PHC clinics to obtain meaningful data. The population of interest consists of parents (guardians or carers) presenting their children (two years old or younger at the time of data collection) to one of the 27 PHC clinics across Qatif.“
-
-
- When a participant received the invitation (link), the first page of the questionnaire included the Personal Information Statement (PIS) to ensure that the participant reads the statement before participating. Then, they would “click to start” to provide their implied consent.
- Statistical analysis: I suggest to insert a measure of the magnitude of the effect for the comparisons. Please consider to include effect sizes.
-
- We carefully considered your comment. However, comparison tests are not within the scope of our work as we are interested in the correlation between a number of variables and childhood immunisation. We noted this in our limitation section and also to motivate future research. The adjusted limitation paragraph reads as follow:
-
-
- “The present study has some limitations. The cross-sectional design of the study limits any causal inferences and the generalisability of the results to the Saudi population. However, the current study is the first empirical investigation of the childhood immunisation in the Eastern Province of Saudi Arabia, the largest province in the kingdom. In addition, a potential recall bias may exist about undocumented data, such as the length of waiting time at PHC centres. It is also possible that some survey questions are under or over-estimated by parents, introducing a non-differential bias. Our data only includes parents attending PHC centres, and this selection bias should be considered during data interpretation. Thus, it is possible that the parents who refused immunisation did not attend the PHC centre and were excluded from this study. The respondents were likely to answer more favourably, introducing a potential social desirability bias. These limitations open avenues for future research and directions aimed at enhancing childhood immunisation, particularly in Saudi Arabia. For instance, future research may examine our model by PHC which may reveal variances across PHC that we could not detect in the current study as our sample would be too small for subgroup analyses. More specifically, future research needs to focus on comparisons between PHCs, place of residence, and other factors and tests whether the size of the detected effects is influenced by these factors.”
-
- Discussion: I also suggest expanding. Emphasize the contribution of the study to the literature in terms of public health.
-
- Thank you for this constructive comment and we do understand that the previous submission did not provide sufficient information and in-depth discussion related to how our work advances the literature. In the revised version, we added a paragraph that discusses our contributions in more details. Here’s what has been added:
-
-
- “Other studies conducted in Saudi Arabia including [11,18-20] have described several factors parents believe that they impact the national childhood immunisation compliance in the kingdom. The limitation with these studies is that they have been limited to parents’ experiences from urban areas of Saudi Arabia. We believed that a more comprehensive examination of these factors was required in both city and rural dwelling families. The major contribution of our study is that we added further investigations to what has been done in the past. Our investigation also included a determination of whether these studied characteristics (e.g., sociodemographic characteristics) differ in more rural communities. Another contribution is that we reviewed relevant literature and tested a number of hypotheses related to the association between demographic, socioeconomic, clinic-related, and parents-related variables and childhood immunisation completion rate. To the best of our knowledge, this is the only study of its kind being conducted in Saudi Arabia and particularly in the Eastern Province. Therefore, this study contributes to the identification of critical factors associated with completion of childhood immunisation and presents important implications to healthcare practitioners, particularly in Saudi Arabia.”
-
Round 2
Reviewer 1 Report
Thank the authors for their excellent work and great responsiveness (in time and quality). The authors reply letter is excellent.
Reviewer 2 Report
All comments have been addressed.
Reviewer 3 Report
the paper was improved and it is now suitable for publication